# The Effects of the PLAYTOD Program on Children’s Physical Activity at Preschool Playgrounds in a Deprived Urban Area: A Randomized Controlled Trial

**DOI:** 10.3390/ijerph17010329

**Published:** 2020-01-03

**Authors:** Nicole Toussaint, Martinette T. Streppel, Sandra Mul, Ruben G. Fukkink, Peter J.M. Weijs, Mirka Janssen

**Affiliations:** 1Faculty of Sports and Nutrition, Center of Expertise Urban Vitality, Amsterdam University of Applied Sciences, Dokter Meurerlaan 8, 1067 SM Amsterdam, The Netherlands; n.toussaint@hva.nl (N.T.); m.t.streppel@hva.nl (M.T.S.); s.mul@hva.nl (S.M.);; 2Faculty of Child Development and Education, Amsterdam University of Applied Sciences, Wibautstraat 2-4, 1091 GM Amsterdam, The Netherlands; r.g.fukkink@hva.nl; 3Faculty of Social and Behavioural Sciences, University of Amsterdam, Nieuwe Achtergracht 127, 1018 WS Amsterdam, The Netherlands; 4Department of Nutrition & Dietetics, Amsterdam University Medical Centers, VU University, De Boelelaan 1117, 1081 HV Amsterdam, The Netherlands

**Keywords:** physical activity, fundamental movement skills, young children, ECEC teachers, preschool, childcare, playground, outdoor environment, intervention, social-ecological model

## Abstract

Interventions to improve children’s physical activity in Early Childhood Education and Care (ECEC) settings are needed. This randomized controlled trial examines the effects of a preschool-based playground program for ECEC teachers in a deprived urban area. On intervention preschools, the PLAYgrounds for TODdlers program (PLAYTOD) was performed. It focused on teacher’s knowledge and skills in order to create a challenging outdoor environment in which young children (2.5 to 4 years old) are able to practice their motor skills. Observations were performed before and after the program with a modified version of the SOPLAY protocol. The activating role of teachers (score from 0 = inactive to 4 = participating), the number of different physical activities, and the quality of children’s physical activity on playgrounds were observed. The latter included the number of performed fundamental movement skills and the estimated physical activity intensity (score from 0 = sedentary to 3 = vigorous). Descriptive statistics and linear regression analyses were used to evaluate the effects of PLAYTOD. After the program, the activating role of teachers on intervention playgrounds improved. Moreover, the program and consecutively the changes made by teachers had a positive effect on the number of different activities and the quality of children’s physical activity. The results emphasize an important role for ECEC teachers in improving physical activity in young children.

## 1. Introduction

The worldwide prevalence of overweight and obesity among young children remains high. According to the World Health Organization, 41 million children aged 0 to 5 years were overweight or obese in 2016 [1]. The problem of excess body weight in children is more severe in deprived urban settings, where it is related to the relatively high number of families with a migration background and/or low socio-economic status [2]. Preschool-aged children show already health inequalities, and the need for early interventions to prevent overweight and obesity in deprived surroundings is widely recognized [3,4,5,6].

Low physical activity levels have been associated with overweight and obesity in young children [7,8,9]. Moreover, recent studies have highlighted the vital importance of motor skills for children’s physical activity levels [10,11]. Children develop motor skills through physical activities and it is in the preschool years that important fundamental movement skills are being evolved. These early learned motor skills form the basis of more complicated movements later in life [12]. Physically active children are shown to have higher motor skills, whereas less active children have lower motor skills [13]. Moreover, children with better motor skills spend more time in moderate-to-vigorous physical activity [14]. A broad foundation of motor skills in the early years predicts a more active lifestyle throughout life and is therefore relevant for the prevention of overweight and obesity [15]. Stodden et al. (2008) presented a conceptual model in which a reciprocal relationship is suggested between physical activity, (perceived) motor competence, physical fitness, and weight status [16]. Interventions in the early years should, therefore, not only focus on physical activity levels but also on stimulating the development of fundamental movement skills.

Childcare settings [17] and their playgrounds are identified as important environments for early interventions to improve physical activity in young children. Ansari et al. (2015) concluded, based on their study among Head Start centers across the U.S. that outdoor preschool play time may serve as an important means to prevent obesity, particularly in low-income children [18]. In Amsterdam, the Netherlands, preschools are important environments to intervene, since parents with a migration background and/or low socio-economic status are advised to enroll their child in preschool (with financial support from the Municipality). These childcare settings provide play-based education up to 15 h per week for young children (2.5 to 4 years old) with mainly disadvantaged backgrounds. Preschools prepare children for primary school, which starts at 4 years of age in the Netherlands [19].

Literature on the involvement of children in physical activity in childcare settings is inconsistent. A recent systematic review of O’Brien et al. (2018) shows that physical activity and sedentary time were highly varied and inconsistent between studies. Despite this variability, the authors conclude that preschool-aged children participate in high rates of sedentary time during childcare [20]. Truelove et al. (2018) draw a similar conclusion in their systematic review on physical activity and sedentary time during childcare outdoor play sessions [21]. The striking high rates of sedentary time during childcare are suggested to contribute to the increasing number of young children with overweight and obesity [22]. These findings from recent studies emphasize the need for interventions to stimulate physical activity in childcare. Finch et al. (2016) conducted a systematic review and meta-analysis on the effectiveness of center-based childcare interventions in improving children’s physical activity. They suggest that the evidence supporting the effectiveness of such interventions is still unclear as mixed results were found by intervention and design characteristics [23]. Furthermore, Mehtälä et al. (2014) conclude in their systematic review that children’s physical activity levels in childcare remained low after intervention studies with a social-ecological approach. They advise future intensive multilevel and multicomponent interventions that pay more attention to the training of Early Childhood Education and Care (ECEC) teachers in promoting physical activity [24]. Wick et al. (2017) conducted a systematic review and meta-analysis on interventions to promote fundamental movement skills in childcare. They implicate that “fundamental movement skills should and can be taught in childcare” [25]. The problem of low motor skills in young children still seems to be urgent. In a recent cross-sectional study by Brian et al. (2019) the developmental delay in motor competence in early childhood is described as “an emerging epidemic” [26]. The outdoor environments at childcare settings provide space and opportunities for challenging physical activities to stimulate fundamental movement skills, which are difficult to perform inside.

Different social-ecological factors in childcare settings predict physical activity behaviors of young children [27,28,29]. The behavior of ECEC teachers is suggested to be important. Teachers can fulfill a pivotal role on playgrounds and focus on the promotion of children’s physical activity and foundation of motor skills. Copeland et al. (2011) describe ECEC teachers as “gatekeepers to the playground” and suggest that individual behaviors and decisions of teachers may result in very different motor experiences of children on the playground, even in the same childcare environments [30].

In light of the pivotal role of ECEC teachers on playgrounds, a preschool-based playground program for ECEC teachers was designed. The multicomponent PLAYgrounds for TODdlers program (PLAYTOD) involves teacher training to improve the quality of physical activity in young children (2.5 to 4 years old) on preschool playgrounds. PLAYTOD is a multicomponent intervention as it addresses both the physical and social environment of young children.

The objective of this study is to examine the effects of PLAYTOD on the behavior of ECEC teachers and consecutively the physical activity behaviors of children on playgrounds in a deprived urban area. The activating role of ECEC teachers, as well as the number of different physical activities and the quality of physical activity in children, was investigated.

## 2. Materials and Methods

### 2.1. Study Design and Setting

This randomized controlled trial was part of the PreSchool@HealthyWeight (PS@HW) study. A detailed description of the PS@HW research design has been published previously [31]. PS@HW was registered in the Netherlands Trial Register (registration number: NL5850) on 26 August 2016. The need for Medical Ethical Approval for the study was waived by The Medical Ethics Review Committee of the VU University Medical Center (reference number: 2016.310). The Committee stated that the Medical Research Involving Human Subjects Act (WMO) does not apply to PS@HW and that an official approval of this study by the Committee is not required. In PS@HW, 41 urban preschools were randomly allocated to an intervention or control group. The randomization was performed by an independent researcher of the Amsterdam University of Applied Sciences with the use of computer-generated randomization lists. The board of the participating childcare organization provided consent for their preschools to participate in this study and observations on their playgrounds. The preschools were mainly located in Amsterdam Nieuw-West, the Netherlands. Amsterdam Nieuw-West is characterized by inhabitants with a migration background and/or low socio-economic status [32]. The prevalence of childhood overweight and obesity in Amsterdam Nieuw-West is high compared to other city districts. In 2017, 12.9% of the children aged 3 years were reported to be overweight or obese in Amsterdam Nieuw-West [33].

The study period for PLAYTOD was on average four months, with observations on playgrounds in the intervention and control group before and after PLAYTOD. Observers were blinded for the experimental condition (intervention or control group) and observation moment (before or after PLAYTOD). Moreover, they were not informed about the content of the program. The PLAYTOD program was carried out only at intervention preschools. Two training sessions were organized by PLAYTOD trainers for in total four training groups of ECEC teachers. In addition, a coaching on the job session was arranged between the training sessions. The program was carried out between March 2017 and November 2017. Table 1 shows an overview of the study period per training group. Figure 1 provides a schematic overview of the PLAYTOD study period.

### 2.2. Intervention

The PLAYTOD program [34] was designed to coach ECEC teachers on how to stimulate physical activity (in particular fundamental movement skills) on the playgrounds of preschools. It focused on knowledge and skills for ECEC teachers in order to create a challenging outdoor environment in which young children (2.5 to 4 years old) are able to practice their motor skills. In the first training session for the intervention training groups, the importance of physical activity for young children was discussed. Moreover, theory about the social-ecological model by Sallis and Owen (1999) was explained. This model emphasizes the relationship between the environment and physical activity behavior. It includes four important levels of influence on physical activity behaviors, namely the policy, environmental, social, and individual level [35,36]. Besides theory on the importance of physical activity and the social-ecological model, a basic inviting structure of the playground was demonstrated and practiced in the first training session. The teachers were taught to create different activity zones on the playground. Specifically, the program taught teachers to implement (1) a cycling zone, (2) a zone with a challenging obstacle course where children were stimulated to jump, climb, crawl, walk, balance and turn, (3) a zone where children were invited to experience different kinds of sensory materials, and (4) a free zone where children were able to run or participate in an organized activity by the ECEC teacher. By organizing the playground in zones with different activities, children were encouraged to practice a variety of fundamental movement skills. In addition, the ECEC teachers were taught to stimulate a “growth mindset” within the children [37]. A “growth mindset” can be achieved by giving positive prompts and using scaffolding to stimulate children to repeat an activity and thereby improve their motor skills. The different activities were organized within the possibilities of the outdoor space and availability of material in preschools. There were no structural changes made to the playground or added material to support the ECEC teachers. Two weeks after the first training session, a trainer of PLAYTOD visited the individual preschools for a coaching on the job session in which specific instructions for improvement were discussed. In the coaching on the job sessions, at least two ECEC teachers of a specific preschool were physically present and an improvement report was written for all intervention preschools. In the second training session for the intervention training groups, there was time to reflect on learning objectives from the first training session. The activating role of ECEC teachers on the playground was practiced and reviewed in more detail.

PLAYTOD was derived from the PLAYgrounds program for primary schools. This physical education-based playground program for primary schools consists of a multicomponent alteration of the schools’ playground. Moreover, the playground usage is stimulated by altered time management of recess times, supervision and encouragement by teachers, and a modification of the physical education content [38]. The initial PLAYgrounds program was proven to be effective in increasing the physical activity intensity levels during recess time in children of 6 to 12 years old [39].

### 2.3. Observations

Per preschool, two observation moments before and two observation moments after PLAYTOD were arranged. Within each observation moment, two observation rounds were performed (resulting in a maximum of four data collection points before and four data collection points after PLAYTOD). Eight (graduate) students of the Amsterdam University of Applied Sciences, degree program Physical Education, were trained to systematically observe in pairs with the use of a modified version of the validated SOPLAY protocol [40]. SOPLAY is a standardized protocol consisting of observations on the quantity of use of the playground in general, type of physical activity, intensity of physical activity, and aspects related to the physical environment (for example, weather conditions and provision of playground equipment). For the purpose of the PLAYTOD program, the SOPLAY observation protocol was adapted to the context of preschools. The observers practiced with a PLAYTOD trainer at different playgrounds to get familiar with the adjusted SOPLAY protocol and the registration of different variables, like fundamental movement skills and physical activity intensity. An interobserver agreement of 92–97% between the different observers and the trainer was obtained after four hours of training. Per observation round, the playgrounds were systematically scanned as a whole for 15 min. Observers were asked to note on preschool level the activating role of ECEC teachers (0 = inactive, no interaction; 1 = inactive, mainly interaction for correction; 2 = enthusiastic encouragement (prompts); 3 = active, active support of game/activity; 4 = participating in game/activity). Furthermore, they counted the number of children and provision of playground equipment on the playgrounds, specified the different activities on the playgrounds, determined the different fundamental movement skills performed by boys and girls separately, and estimated the physical activity intensity for boys and girls separately (0 = sedentary, i.e., sitting and not involved in an activity or sitting and slowly scooping in the sandbox; 1 = light, i.e., standing or walking slowly or being at moderate pace on the balance bike or scooping in the sandbox while standing; 2 = moderate, i.e., walking, crawling, jumping from height, going up and down on the slide, throwing or passing a ball, or being at high pace on the balance bike; 3 = vigorous, i.e., running or jumping for more than 20 s). In this study, the quality of physical activity was defined as the number of performed fundamental movement skills and the estimated physical activity intensity on playgrounds.

Additional observations at intervention preschools were performed in June 2019 (±1.5 years after PLAYTOD). These retention observations intended to evaluate the long-term use of learned skills by ECEC teachers in the intervention group. The ECEC teachers were mainly the same as during the first data collection period (per preschool, at least one trained teacher of the first observation period was present at the second observation period). The groups of children were different due to changes in the group compositions (children turned 4 and went to primary school)). Further, new (blinded) observers were trained and the same observation protocol was used. An interobserver agreement of 94–96% between the different observers and the trainer was obtained after four hours of training. Again, two observation moments, with two observation rounds each, were arranged per preschool (resulting in a maximum of four waves of data collection). Similar to the first data collection period, the activating role of ECEC teachers, the different activities, the performed fundamental movement skills, and the physical activity intensity on playgrounds were observed.

### 2.4. Statistical Analyses

The statistical analyses were performed with IBM SPSS Statistics 25 (IBM corp., Armonk, NY, USA). Data was collected and analyzed on preschool level. Descriptive statistics (mean ± SD) were used to present the observed values for the outcome measures on each data collection point before and after PLAYTOD. In addition, linear regression analyses were performed to evaluate the effects of PLAYTOD. Per outcome measure, the data collection points after PLAYTOD were averaged and the obtained overall mean values were used as dependent variables. Experimental condition (intervention or control group), averaged values of the data collection points before PLAYTOD, and dummy variables for the training groups were used as independent variables. A two-tailed statistical significance level of *p* < 0.05 was used in all statistical analyses. Data collection points of the retention observations were averaged and one overall mean value per outcome measure (mean ± SD) was presented.

## 3. Results

### 3.1. Study Sample

In the PS@HW trial, 41 preschools were randomized of which 21 preschools were allocated to receive the intervention. However, two preschools in the intervention group did not take part in the PLAYTOD program as it was considered to put too much strain on the teachers at these locations by the management. Furthermore, two preschools in the intervention group and one preschool in the control group were excluded from analyses (resulting in 17 analyzed intervention preschools and 19 analyzed control preschools). Observations at these locations were not possible due to renovation work at the playground or school building. Figure 2 shows a flow diagram of the study sample. Not all four observation moments were performed at each included preschool because of time constraints or weather conditions. However, on all playgrounds at least one observation moment before and one observation moment after PLAYTOD was conducted.

In total, 14 of the 17 initial analyzed intervention preschools were available for the retention observations. At least one observation moment took place at each of these preschools.

### 3.2. Intervention Effects

Figure 3 shows the observed values (mean ± SD) for the outcome measures on each data collection point before and after PLAYTOD. Before the program, an inactive role of ECEC teachers was observed on all playgrounds. After the program, the activating roles ‘enthusiastic encouragement (prompts)’, ‘active support of activity’, and ‘participating in activity’ were observed on intervention playgrounds, whereas on control playgrounds inactive roles by teachers were still observed. Moreover, on intervention playgrounds new activities (obstacle course, ball game, and guided physical activity) were introduced by the ECEC teachers and the number of performed fundamental movement skills in boys and girls sextupled. After PLAYTOD, new fundamental movement skills like jumping, balancing, throwing, rolling, crawling, and turning were observed on intervention playgrounds, whereas this was not the case on control playgrounds. Furthermore, the mean estimated physical activity intensity in boys and girls on each data collection point improved from sedentary-to-light levels to light-to-moderate levels on intervention playgrounds. No changes in physical activity intensity were observed on control playgrounds.

Table 2 presents the averaged observed values of the data collection points before and after PLAYTOD and the results of the linear regression analyses. A positive intervention effect was found for the activating role of ECEC teachers on the playgrounds. After PLAYTOD, the score for the activating role of ECEC teachers in the intervention group increased with 2 points compared to the control group (β = 2.02, 95% CI = 1.85; 2.20). Moreover, the program had a positive effect on the number of different activities (β = 1.29, 95% CI = 0.78; 1.80). The same applies to the number of fundamental movement skills performed by boys (β = 4.74, 95% CI = 4.19; 5.29) and girls (β = 5.29, 95% CI = 5.04; 5.53). In addition, a positive intervention effect was found on the estimated physical activity intensity shown by boys (β = 0.71, 95% CI = 0.40; 1.02) and girls (β = 0.87, 95% CI = 0.47; 1.27). For both the intervention and control group, there were on average ten children (mean ranged from 9.7–10.4; SD ranged from 2.6–3.1) on the playgrounds. In addition, the mean score for the number of provision of playground material ranged from 3.6–4.0 (SD ranged from 0.7–1.0) for both experimental groups.

At the retention observations, an averaged observed value of 1.74 ± 0.38 was found for the activating role of ECEC teachers. Moreover, on average there were 4.00 ± 0.73 different activities on the playgrounds. The averaged value of performed fundamental movement skills was 4.80 ± 0.69 for boys and 3.84 ± 0.92 for girls. The averaged estimated physical activity intensity was 1.39 ± 0.53 for boys and 1.29 ± 0.41 for girls.

## 4. Discussion

This study investigated the effects of the preschool-based playground program PLAYTOD for ECEC teachers in improving the quality of children’s physical activity on playgrounds of Dutch preschools. The program was effective in improving the teachers’ activating role, the number of different physical activities, and the quality of children’s physical activity on playgrounds. The study was conducted in a deprived urban area with a high prevalence of childhood overweight and obesity.

An important finding of this study was the improved activating role of ECEC teachers on the playgrounds. After the program, ECEC teachers from the intervention group created a challenging outdoor environment and gave enthusiastic encouragements (prompts) or active support of activities. Furthermore, they sometimes participated in physical activities. The pivotal role for ECEC teachers on playgrounds has been suggested [30]. However, evidence for the promotion of physical activity by ECEC teachers is lacking [41]. Our results support the important role of ECEC teachers on playgrounds.

The program and consecutively the changes made by ECEC teachers during outdoor play sessions, improved the quality of children’s physical activity on playgrounds. In the intervention group, the number of performed fundamental movement skills and estimated physical activity intensity on playgrounds increased. The observed gains are relevant because many young children spend a large part of their time in sedentary behavior and have low motor competence [20,21,26]. Results of the MAMBO cohort in Amsterdam, the Netherlands, emphasize the problem of low motor competence in urban settings. In 2017, 16% of the children in this cohort (6 to 12 years old) were reported to have a developmental delay in motor competence [42]. In addition, many preschool-aged children do not meet physical activity recommendations [43]. Guidelines for physical activity in young children differ between countries and various countries, including the Netherlands, do not have specific recommendations for young children yet. In recently published movement guidelines for preschool-aged children, recommendations are formulated for variety, duration, and physical activity intensity. For example, the Canadian 24 h Movement Guidelines for the Early Years recommends preschoolers to be at least 180 min physically active in a variety of physical activities spread throughout the day, of which at least 60 min is moderate-to-vigorous physical activity [44]. PLAYTOD may contribute to achieving physical activity recommendations, in particular when it concerns the variety of physical activities.

In this study, the estimated physical activity intensity on playgrounds improved, but did not reach the moderate-to-vigorous level. In contrast, Pate et al. (2016) reported that a flexible ecologic physical activity intervention at preschools increased moderate-to-vigorous physical activity in 4-year-olds. The intervention described by Pate et al. (2016) did not only focus on physical activity on playgrounds, but trained teachers to provide children with opportunities to be active throughout the school day. Moreover, the study of Pate et al. measured physical activity on an individual level with accelerometers and not via direct observations of a group of children as in our study [45]. PLAYTOD focused more on the development of fundamental movement skills rather than physical activity intensity or duration. When considering the age category of 2.5 to 4 years old we think it should be highlighted that our program was successful in improving the number of performed fundamental movement skills. By organizing the playground in different activity zones, the children were able to practice a variety of fundamental movement skills. This is important because a better motor skill development in the early years is associated with higher motivation for participation in physical activities, which may contribute to an active lifestyle throughout life and the prevention of overweight and obesity [11,15,16].

Strengths of this study include the use of a randomized controlled design and the multicomponent character of PLAYTOD. The program did neither make structural changes to the playgrounds nor added supporting materials for ECEC teachers. The different activities were organized within the possibilities of the existing outdoor space and availability of materials in the preschools. This makes the program feasible to implement in a variety of ECEC settings.

Limitations include the ability to generalize the findings. Only preschools of one large childcare organization in Amsterdam were included. Furthermore, the fact that ECEC teachers knew when observations took place may be a limitation for the evaluated effects of the program. It could be the case that teachers put more effort in their role on the playground only when observers were present. The use of less obtrusive methods for data collection at teacher level (e.g., unplanned video recordings) may therefore be of use in future trials. In the current study, only data on preschool level was obtained via direct observations. In future trials, accelerometers could be used to obtain additional objectively measured data on children’s individual physical activity intensity and duration. For individual data on fundamental movement skills, separate direct observations per child could be performed. Lastly, no long-term effects of the program were investigated in this study. However, we performed additional observations at the intervention preschools in June 2019 (±1.5 years after PLAYTOD) as part of a retention measure. Although the score for the activating role of teachers, the number of performed fundamental movement skills, and the estimated physical activity intensity were slightly lower at the retention observations compared to scores right after the performance of PLAYTOD, an improvement for all outcome measures was still observed compared to data before the performance of PLAYTOD. In future studies, it could be interesting to add a technological tool for ECEC teachers to help them recognize real-time low physical activity. Byun et al. (2018) recently published a study with a promising application of real-time physical activity monitoring by teachers and teacher-regulated strategies to promote physical activity in preschool-aged children [46].

## 5. Conclusions

The preschool-based playground program PLAYTOD improved the quality of children’s physical activity on the playgrounds of urban preschools in a deprived area. Our results emphasize the important role of ECEC teachers in stimulating physical activity in young children. The program contributes to the professional development of ECEC teachers and addresses the call for early interventions to prevent overweight and obesity.

## Figures and Tables

**Figure 1 ijerph-17-00329-f001:**
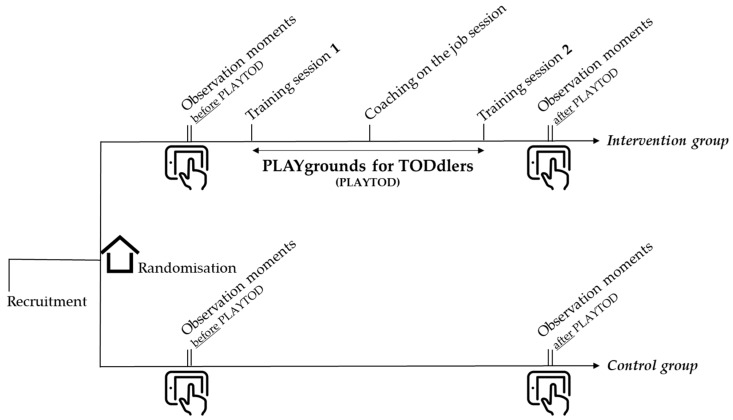
Schematic overview of the study.

**Figure 2 ijerph-17-00329-f002:**
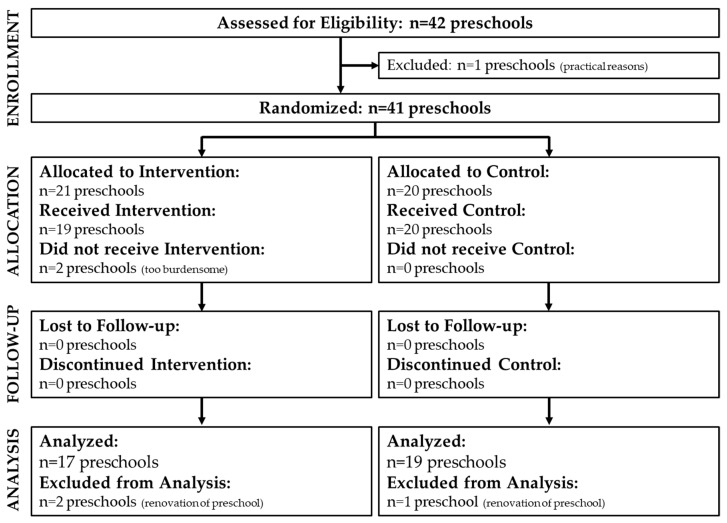
Flow diagram of the study sample.

**Figure 3 ijerph-17-00329-f003:**
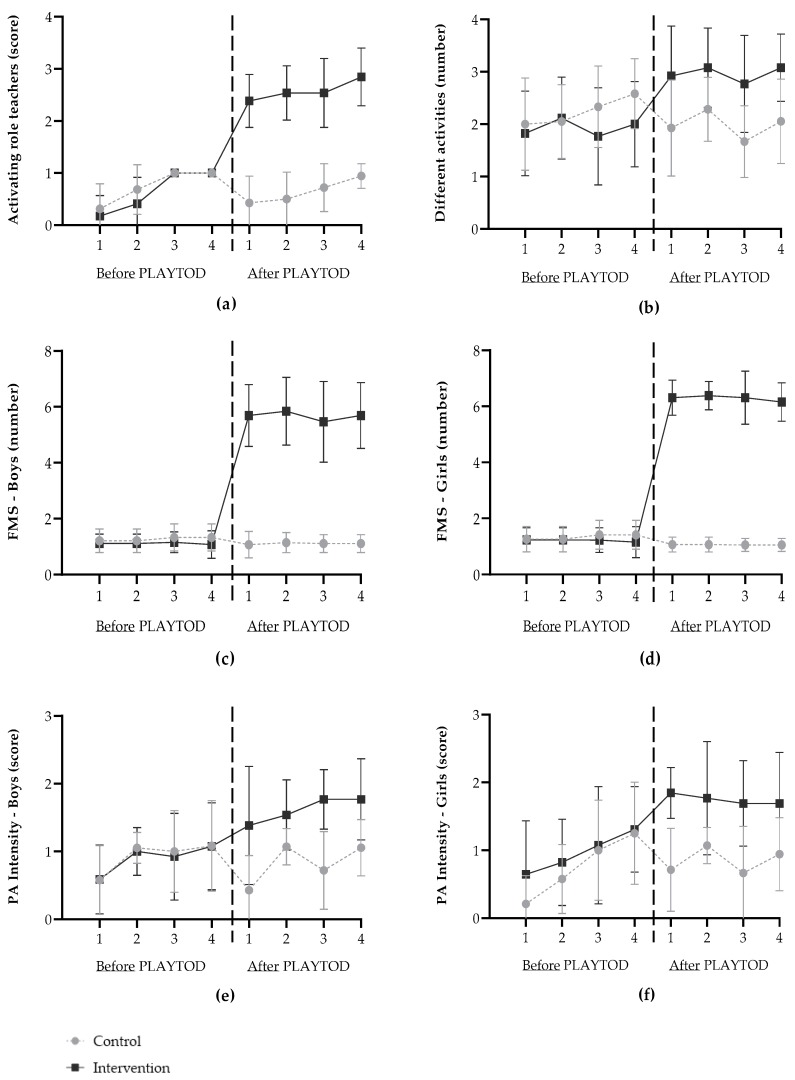
Observed values (mean ± SD) on each data collection point for (**a**) the activating role of Early Childhood Education and Care teachers: original categories were 0 = inactive (no interaction), 1 = inactive (mainly interaction for correction), 2 = enthusiastic encouragement (prompts), 3 = active (active support of activity), 4 = participating in activity; (**b**) the number of different activities on the playground; (**c**) the number of Fundamental Movement Skills (FMS) performed by boys; (**d**) the number of Fundamental Movement Skills (FMS) performed by girls; (**e**) the estimated physical activity (PA) intensity in boys: original categories were 0 = sedentary, 1 = light, 2 = moderate, 3 = vigorous; (**f**) the estimated physical activity (PA) intensity in girls: original categories were 0 = sedentary, 1 = light, 2 = moderate, 3 = vigorous.

**Table 1 ijerph-17-00329-t001:** Overview of the study period (March–November 2017) per training group.

	Observations before PLAYTOD	PLAYTOD Sessions	Observations after PLAYTOD
Training group 1	March	April–June	July
Training group 2	May	June–September	October
Training group 3	September	September–November	November
Training group 4	September	September–November	November

PLAYTOD: PLAYgrounds for TODdlers program.

**Table 2 ijerph-17-00329-t002:** Averaged observed values of the data collection points before and after PLAYTOD and the results of the linear regression analyses.

	Averaged Observed Values (Mean ± SD)				Results Linear Regression Analyses ^1^			
	Intervention (*n* = 17)		Control (*n* = 19)		β	*p*	95% CI	
	Before	After	Before	After			LCL	UCL
Activating role ^2^	0.59 ± 0.20	2.59 ± 0.32	0.66 ± 0.24	0.66 ± 0.24	2.02	<0.001	1.85	2.20
No. Activities ^3^	1.91 ± 0.75	3.06 ± 0.72	2.05 ± 0.76	1.89 ± 0.70	1.29	<0.001	0.78	1.80
No. FMS ^4^								
Boys	1.10 ± 0.34	5.59 ± 1.15	1.21 ± 0.42	1.08 ± 0.34	4.74	<0.001	4.19	5.29
Girls	1.22 ± 0.45	6.28 ± 0.46	1.26 ± 0.45	1.05 ± 0.23	5.29	<0.001	5.04	5.53
PA intensity ^5^								
Boys	0.85 ± 0.34	1.57 ± 0.54	0.88 ± 0.28	0.84 ± 0.27	0.71	<0.001	0.40	1.02
Girls	0.91 ± 0.50	1.75 ± 0.59	0.67 ± 0.28	0.87 ± 0.41	0.87	<0.001	0.47	1.27

PLAYTOD: PLAYgrounds for TODdlers program. LCL: Lower Confidence limit. UCL: Upper Confidence Limit. ^1^ Regression models were adjusted for the baseline value of the outcome variable and the training groups. ^2^ The activating role of Early Childhood Education and Care teachers: original categories were 0 = inactive (no interaction), 1 = inactive (mainly interaction for correction), 2 = enthusiastic encouragement (prompts), 3 = active (active support of activity), 4 = participating in activity. ^3^ The number of different activities on the playground. ^4^ The number of Fundamental Movement Skills (FMS) performed by boys and girls. ^5^ The estimated physical activity (PA) intensity in boys and girls: original categories were 0 = sedentary, 1 = light, 2 = moderate, 3 = vigorous.

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
