# Peer review of "The Effects of the PLAYTOD Program on Children’s Physical Activity at Preschool Playgrounds in a Deprived Urban Area: A Randomized Controlled Trial"

_ijerph, 2020, doi:10.3390/ijerph17010329_

Round 1
Reviewer 1 Report
REVIEW_ IJERPH _666967
Title: Effectiveness of the PLAYTOD Program on 
Children’s Physical Activity at Preschool 
Playgrounds in a Deprived Urban Area: A Randomized Controlled Trial 

General view
I really think this study is very interesting. It brings knowledge that can be applied in other contexts, anywhere in the world. Movement professionals, educators, and researchers can benefit from the content of this article. There are weaknesses in the article, but they can be properly corrected.
I hope I have contributed to make the article even more accurate and clear. And thank you for the opportunity to make this review.
Title: The title brings the term effectiveness. But the study, in fact, measures the effect of an intervention program. I suggest that the authors adjust the title so that it is accurate for the purpose of the study.
Abstract
The Abstract should be reviewed from the comments below on the concept of Proximal Development Zones.
Introduction
The Introduction is good, focusing on the results of systematic reviews on the focus of the article. However, there is a need to incorporate an important recently published article on interventions in preschool children, which has not been reviewed in such reviews. (Byun, Lau, and Brusseau 2018)
Byun, Wonwoo, Erica Y. Lau, and Timothy A. Brusseau. 2018. “Feasibility and Effectiveness of a Wearable Technology-Based Physical Activity Intervention in Preschoolers: A Pilot Study.” International Journal of Environmental Research and Public Health 15 (9). https://doi.org/10.3390/ijerph15091821.
The wording of the objective must be adjusted. As it is written, it is not clear whether the program was designed to have an effect on the behavior of children's activities, the behavior of the teachers who mediate these activities, or both children and teachers. Lines 106 and 107 - The phrase "the quality of physical activity was defined as the number of performed fundamental movement skills and the estimated physical activity intensity on playgrounds" should be moved to the methods section.
Method
The research design is appropriate to objective. The methods needs some review.What seems to me to need a more careful review and / or description is about the Proximal Development Zone; This is a concept originated in the works of Lev Vygostky, and was used without being referenced in the Abstract and subsection 2 Intervention. In fact, this concept is very weakly placed in the present study. Thus, it is questioned whether the authors used this concept or not. If this was the case - they used Vygotisky's concept (Vygostky 1986) - this should be better clarified in the manuscript.Vygostky, Lev Semenovich. 1986. Thought and Language. Vol. 5. Cambridge: MIT Press. https://doi.org/10.3233/BEN-1992-5106.
Also, it can be seen that the authors use the term "zone" perhaps to identify the places marked / intended to carry out the different activities. The reader may be confused by this interchangeable use of words. To preserve the accuracy of a scientific text, I suggest that the term "zone" be replaced by another without losing its descriptive power. It was not possible to understand how observers estimated physical activity intensity (0 = sedentary; 1 = light; 2 = moderate; 3 = vigorous). This needs to be better explained.
Results
n the graphics, it is a bit confusing to identify the variables. The Y axis should indicate the variable name and the unit of measure (number / points / scores). On the X axis there is a lot of unnecessary information. It should only be indicated "before PLAYTOD" and "after PLAYTOD"In the title of the graphs it should appear that the mean and standard deviation are described in the graphs and not in the Y axis.
Discussion and conclusion
Lines 320/322 - The reference used to support the statements can be reinforced by a series of studies on motor competence and the perception of motor competence. It would be desirable for the authors to review the articles by Stodden et al. (2008) and Robinson et al. (2015)
Robinson, Leah E., David F. Stodden, Lisa M. Barnett, Vitor P. Lopes, Samuel W. Logan, Luis Paulo Rodrigues, and Eva D’Hondt. 2015. “Motor Competence and Its Effect on Positive Developmental Trajectories of Health.” Sports Medicine 45 (9): 1273–84. https://doi.org/10.1007/s40279-015-0351-6.
Stodden, David F., Jacqueline D. Goodway, Stephen J. Langendorfer, Mary Ann Roberton, Mary E. Rudisill, Clersida Garcia, and Luis E. Garcia. 2008. “A Developmental Perspective on the Role of Motor Skill Competence in Physical Activity: An Emergent Relationship.” Quest 60 (2): 290–306. https://doi.org/10.1080/00336297.2008.10483582.
Reviewer 2 Report
The study emphasize the important role of ECEC teachers in stimulating physical activity (PA) in young children. As it was proved the preschool-based playground programs may improve the quality of children’s PA on the playgrounds even of urban areas. This is obvious nova days that a professional development of ECEC teachers’ is required in terms of knowledge and at the same time skills regarding PA (in the early years of children mostly undertaken through play as spontaneously/freely taken physical movement). This a not only for the overweight or obesity reason. It is essential for the holistic child development (i.e. motor, cognitive and social skills). In this regard, this study and paper seems to be very important for the sound development of children and to the future society.
But as the Reviewer I have some comments and suggestions to the Authors:
Line 50-51, p.2.: “Physically active children are shown to have higher motor skills, whereas less active children have lower motor skills”, it needs references. Line 215, p.5: “Forty-one” and then “21”, please standardise the wording in whole text (L.224). In a Figure 2. There are 42 preschool and then in the text there are 41? The same is with interviewed preschools: L. 22, p:1: “n=19” and then Fig. 2 “17”? Line 316, p. 9.: “Our study focussed MORE on the development of fundamental movement skills rather than physical activity intensity or duration”. And my question is: Which preschools were selected for the second observation? Were the same as for the first intervention, or there were different? Thus, were the groups of children the same (but after 1,5 years) or different and the teachers were the same? The reference which might be used: Katarzyna OstrzyĹĽek-PrzeĹşdziecka, Cynthia Smeding, MichaĹ‚ Bronikowski, Mariusz Panczyk, Wojciech Feleszko. „The association of physical activity and sedentary behaviors with upper respiratory tract infections and sleep duration in preschool children - study protocol, International Journal of Environmental Research and Public Health 2019:16(9 [1496]), p-ISSN: 1660-4601, DOI: 10.3390/ijerph16091496;Arya Ansari, M.A., Kierra Pettit, B.S., Elizabeth Gershoff, Combating Obesity in Head Start: Outdoor Play and Change in Children’s BMI, J Dev Behav Pediatr. 2015 Oct; 36(8): 605–612, doi:10.1097/DBP.0000000000000215;
After above mentioned suggestions and comments I guess this article will be worth to published in regard of one’s awareness in term of PA in the early age of humankind.
Reviewer 3 Report
It is an interesting study targeting physical inactivity in preschool children.
Please see below some points for improvements:
p.1 l. 26-29: Please restructure this part as currently it is unclear and confusing.
p.2 l .62. Why is preschool important only for low migration/SES population. In this paragraph, it is implied that only disadvantaged children are entitled or attending preschool. Is this correct?
Earlier in the introduction you talked about the increasing levels of child obesity/ overweight children
p.3 l. 3-7: Again, this paragraph needs to be re-written and in order to outline the research aims more explicitly.
p.3 l. 124: Did observers receive any training?
Did participants give written consent form?
p.4 l.141: if you refer to the ZPD, you need to cite Vygotsky’s theories
You add a table on the main points developed during training sessions derived from the theoretical frameworks used
p.4 l.161: did you conduct any fidelity checks for evaluation the intervention components?
p.5 l. 181: Add information about the validity of SOPLAY
p.5 l. 184 please provide more information about the FMS assessment. Were children videotaped?
Were there any scales used for evaluating FMS (e.g., Barnett)?
p.5 l.190-193: were the PA observations conducted per individual or per group. How children were classified for 15 min. It is impossible that they can be engaged in VPA for that long time.
Also, each observer was observing one child or more?
p.5 l.196: were new inter-observer reliability scores conducted for the new blinded observers? How did authors ensure the consistency among observers?
p.9 l. 309-313. I am bit doubtful on how PA intensity was accurately measured through observations. This methodological flaw can be noted as a limitation of this study.
Also, more information is required on how FMS were assessed and resulted in improved outcomes
p.9 l. 331: you can also refer to “priming effects” as well as possible expectation biases occurred.
Overall, the assessments tools currently used would be more informative of adherence to the intervention components. However, more objective, valid and reliable measures could have increased the quality of this study.
Please check all the references, there are some typos noticed.
Author Response
Reviewer 3
It is an interesting study targeting physical inactivity in preschool children. Please see below some points for improvements:
Point 1:
p.1 l. 26-29: Please restructure this part as currently it is unclear and confusing.
Response 1:
We thank the reviewer for the comment and restructured this part in the abstract: “The activating role of teachers (score from 0=inactive to 4=participating), the number of different physical activities and the quality of children’s physical activity on playgrounds were observed. The latter included the number of performed fundamental movement skills and the estimated physical activity intensity (score from 0=sedentary to 3=vigorous).”. (lines 27-31 revised manuscript with tracked changes)
Point 2:
p.2 l .62. Why is preschool important only for low migration/SES population. In this paragraph, it is implied that only disadvantaged children are entitled or attending preschool. Is this correct?
Earlier in the introduction you talked about the increasing levels of child obesity/ overweight children
Response 2:
All children can attend the preschools. However, in Amsterdam (the Netherlands), particularly parents with a migration background and/or low socio-economic status are advised (by the Municipality) to enroll their child in preschool (these families also receive financial support from the Municipality). So, the children in preschools are mainly from families with a migration background and/or low socio-economic status. We rephrased the sentences: “In Amsterdam, the Netherlands, preschools are important environments to intervene, since parents with a migration background and/or low socio-economic status are advised to enroll their child in preschool (with financial support from the Municipality). These childcare settings provide play-based education up to 15 hours per week for young children (2.5 to 4 years old) with mainly disadvantaged backgrounds. Preschools prepare children for primary school, which starts at 4 years of age in the Netherlands.”. (lines 65-70)
Point 3:
p.3 l. 3-7: Again, this paragraph needs to be re-written and in order to outline the research aims more explicitly.
Response 3:
We think the reviewer meant p.3 l.97-107 of version 1 and acknowledge the problem. We restructured the paragraphs: “In light of the pivotal role of ECEC teachers on playgrounds, a preschool-based playground program for ECEC teachers was designed. The multicomponent PLAYgrounds for TODdlers program (PLAYTOD) involves teacher training to improve the quality of physical activity in young children (2.5 to 4 years old) on preschool playgrounds. PLAYTOD is a multicomponent intervention as it addresses both the physical and social environment of young children.
The objective of this study is to examine the effects of PLAYTOD on the behavior of ECEC teachers and consecutively the physical activity behaviors of children on playgrounds in a deprived urban area. The activating role of ECEC teachers as well as the number of different physical activities and the quality of physical activity in children were investigated.”. (lines 101-110)
Point 4:
p.3 l. 124: Did observers receive any training? Did participants give written consent form?
Response 4:
We thank the reviewer for the comments. The observers did receive training, please see section 2.3. Observations: “The observers practiced with a PLAYTOD trainer at different playgrounds to get familiar with the adjusted SOPLAY protocol and the registration of different variables, like fundamental movement skills and physical activity intensity. An interobserver agreement of 92-97% between the different observers and the trainer was obtained after four hours of training.” (lines 192-196) In addition, we added a statement about consent to the manuscript: “The board of the participating childcare organization provided consent for their preschools to participate in this study and observations on their playgrounds.”. (lines 125-126)
Point 5:
p.4 l.141: if you refer to the ZPD, you need to cite Vygotsky’s theories
Response 5:
We acknowledge the problem and think it is better to not use the concept by Vygostky in our manuscript. Rephrased sentence: “It focussed on ECEC teacher’s knowledge and skills in order to create a challenging outdoor environment in which young children (2.5 to 4 years old) are able to practice their motor skills”. (lines 23-25 & 147-149)
Point 6:
You add a table on the main points developed during training sessions derived from the theoretical frameworks used.
Response 6:
We thank the reviewer for the suggestion to add a table but prefer to describe the theoretical frameworks in the text (for space and readability purposes).
Point 7:
p.4 l.161: did you conduct any fidelity checks for evaluation the intervention components?
Response 7:
We thank the reviewer for the comment. During the observations, the different activities (organized by teachers) on the playgrounds and the activating role of teachers were observed. As suggested by our results, the teachers on intervention preschools were able to provide a challenging outdoor environment for the children, as meant by the PLAYTOD training.
Point 8:
p.5 l. 181: Add information about the validity of SOPLAY
Response 8:
We agree with the reviewer and added to the manuscript that SOPLAY is a validated observation protocol. (line 188)
Point 9:
p.5 l. 184 please provide more information about the FMS assessment. Were children videotaped?
Were there any scales used for evaluating FMS (e.g., Barnett)?
Response 9:
In this study, observers determined the different FMS performed by boys and girls/number of FMS performed by boys and girls. For example, if the FMS jumping, balancing and walking were observed in girls, the number of performed FMS in girls was 3. It is all about variation in FMS, no scales were used to evaluate FMS.
Point 10:
p.5 l.190-193: were the PA observations conducted per individual or per group. How children were classified for 15 min. It is impossible that they can be engaged in VPA for that long time.
Also, each observer was observing one child or more?
Response 10:
The observations were performed on preschool/group level. Per observation round, a playground was systematically scanned as a whole. So, observers did not follow individual children for 15 minutes but determined for example all different FMS performed by boys and girls during an observation period of 15 minutes. This was repeated for the next 15 minutes (both 15 minute rounds make one observation moment). Following the SOPLAY protocol, the average group activity level for that observation moment was estimated. With the observation moments before and observation moments after the intervention, we were able to evaluate the effects of the intervention. We added the words “systematically scanned” to line 197.
Point 11:
p.5 l.196: were new inter-observer reliability scores conducted for the new blinded observers? How did authors ensure the consistency among observers?
Response 11:
We thank the reviewer for the comment. The interobserver agreement was determined in the same way as during the pre and post observations. Added sentence: “An interobserver agreement of 94-96% between the different observers and the trainer was obtained after four hours of training”. (lines 217-219)
Point 12:
p.9 l. 309-313. I am bit doubtful on how PA intensity was accurately measured through observations. This methodological flaw can be noted as a limitation of this study.
Response 12:
Observers did not measure physical activity but estimated the physical activity intensity based on performed activities. We added some examples to the text to clarify how physical activity intensity was estimated: “(0 = sedentary, i.e. sitting and not involved in an activity or sitting and slowly scooping in the sandbox; 1 = light, i.e. standing or walking slowly or being at moderate pace on the balance bike or scooping in the sandbox while standing; 2 = moderate, i.e. walking, crawling, jumping from height, going up and down on the slide, throwing or passing a ball or being at high pace on the balance bike; 3 = vigorous, i.e. running or jumping for more than 20 seconds)”. (lines 203-208)
Point 13:
Also, more information is required on how FMS were assessed and resulted in improved outcomes
Response 13:
In this study, observers determined the different FMS performed by boys and girls/number of FMS performed by boys and girls. For example, if the FMS jumping, balancing and walking were observed in girls, the number of performed FMS in girls was 3. It is all about variation in FMS, no scales were used to evaluate FMS.
Point 14:
p.9 l. 331: you can also refer to “priming effects” as well as possible expectation biases occurred.
Response 14:
We think the reviewer comments on the behavior of the ECEC teachers when being observed. We reflected on this limitation in the discussion. On the other hand, teachers in the control group were also observed. Both groups might have shown some overestimation of their role and consecutively the effects on the physical activity behavior of children. Still, we found a significant difference between groups.
Overall, the assessments tools currently used would be more informative of adherence to the intervention components. However, more objective, valid and reliable measures could have increased the quality of this study.
Point 15:
Please check all the references, there are some typos noticed.
Response 15:
We thank the reviewer for this comment and corrected the typos.
